# Salicin and Hederacoside C-Based Extracts and UV-Absorbers Co-Loaded into Bioactive Lipid Nanocarriers with Promoted Skin Antiaging and Hydrating Efficacy

**DOI:** 10.3390/nano12142362

**Published:** 2022-07-10

**Authors:** Ioana Lacatusu, Brindusa Balanuca, Andrada Serafim, Cristina Ott, Mariana Prodana, Nicoleta Badea

**Affiliations:** 1Department of Organic Chemistry, Faculty of Chemical Engineering and Biotechnologies, University POLITEHNICA of Bucharest, Polizu No 1, 011061 Bucharest, Romania; ioana.lacatusu@upb.ro (I.L.); brindusa.balanuca@upb.ro (B.B.); cristina.ott@upb.ro (C.O.); 2Advanced Polymer Materials Group, Faculty of Chemical Engineering and Biotechnologies, University POLITEHNICA of Bucharest, Polizu No 1, 011061 Bucharest, Romania; andrada.serafim0810@upb.ro; 3Center for Surface Science and Nanotechnology (CSSNT), University POLITEHNICA of Bucharest, Splaiul Independentei, no. 313, 060042 Bucharest, Romania; mariana.prodana@upb.ro; 4Department of General Chemistry, Faculty of Chemical Engineering and Biotechnologies, University POLITEHNICA of Bucharest, Polizu No 1, 011061 Bucharest, Romania

**Keywords:** lipid nanostructured hydrogels, UVA and UVB filters, photoprotection, photostability, skin health promoting extracts

## Abstract

Conventional and herbal active principles can be combined in a beneficial harmony using their best features and compensating for the certain weaknesses of each. The study will answer the question, “how can willow bark extract (*Wbe*) or ivy leaf extract (*Ile*) influence the photoprotective, skin permeation and hydration properties of Bioactive Lipid Nanocarriers (BLN) loaded with UV-filters and selected herbals?”. BLN-*Wbe/Ile*-UV-filters were characterized for particle size, zeta potential, thermal behavior, entrapment efficiency and drug loading. The formulated BLN-hydrogels (HG) were subjected to *in vitro* release and permeation experiments. The *in vitro* determination of sun protection factors, as well as comparative *in vitro* photostability tests, rheology behavior and *in vivo* hydration status have been also considered for hydrogels containing BLN-*Ile*/*Wbe*-UV-filters. Photoprotection of BLN-HG against UVA rays was more pronounced as compared with the UVB (UVA-PF reached values of 30, while the maximum SPF value was 13). The *in vitro* irradiation study demonstrated the photostability of BLN-HG under UV exposure. A noteworthy cosmetic efficacy was detected by *in vivo* skin test (hydration effect reached 97% for the BLN-*Wbe*-UV-filters prepared with pomegranate oil). The research novelty, represented by the first-time co-optation of the active herbal extracts (*Wbe* and *Ile*) together with two synthetic filters in the same nanostructured delivery system, will provide appropriate scientific support for the cosmetic industry to design novel marketed formulations with improved quality and health benefices.

## 1. Introduction

Skin provides a large surface area for the topical delivery of therapeutic formulations intended for skin protection against complex skin diseases such as skin cancers [1]. The pathogenesis of skin cancer is multi-factorial, but one of the prevalent risk factors is long exposure to UV radiation, which causes the appearance of skin photocarcinogenicity, due to impairment of genetic material, activating tumor promoter genes, inflammation, and oxidative stress [2,3]. Because of the detrimental UV radiations effects, photoprotection—as an essential prophylactic and therapeutic element–represents a mandatory necessity to avoid these undesired effects [4]. The quality of photoprotection provided by sunscreen products have improved considerably in time. The most efficient sunscreen products should provide large-spectrum UV protection, offering a broad protection against both types of UVB and UVA radiations. For this reason, the commercial sunscreens typically contain at least three UV filters in concentration ranging between 3% and 10%, one with optimal performance in the UVA region and the other one in the UVB region. Their efficacy in reducing skin photo carcinogenesis and photo ageing is widely documented [5]. However, there are many issues about UV filter safety that have raised significant concerns in recent years due to their continuous usage, persistent input, and potential threat to human health [6]. The presence of high amounts of UV filters which usually leads to synergistic effects regarding both, the photoprotective performance and photostabilization action of the UV-filters, could lead to an acceleration of their decomposition and to several undesirable actions [7,8].

The main issues linked to the UV filters are their considerable concentrations added in the marketed sunscreen products, i.e., 5–10% and aspects related to the poor stability of several UV filters at UV photons action [9]. The potential side effects generated using photoprotective formulations must be analyzed from several perspectives: (i) The poor stability of organic sunscreens which can be transformed into reactive intermediates under the action of light: once affected, these compounds have no longer any blocking effect of UV radiation, and moreover, they can exhibit a harmful action after the reaction with UV [10,11]; (ii) the penetration of sunscreens in the upper layers of the skin and subsequently in the blood circulation: this systemic absorption is the most dangerous, as the lipophilic nature of many UV filters can cause body bioaccumulation [12]; (iii) occurrence of allergic reactions: among the organic UV absorbers, octocrylene, benzophenone-3 and avobenzone, the most common sunscreens used in photoprotection products, frequently elicited photoallergic contact dermatitis [13,14]; (iv) other concerns associated with disturbance of hormonal activity caused by UV filters, e.g., certain filters activated estrogen receptors [6,15,16].

As a response to these issues, an alternative approach for photoprotection in the topical application is the simultaneous use of low amounts of UV-filters with natural antioxidants (e.g., vitamins A, C, and E) and/or phytochemicals (e.g., vegetable oils and herbal extracts) enriched in flavonoids and polyphenols which could maintain or restore a healthy skin barrier [17,18]. Nowadays, there is an increasing interest for the “natural chemicals” or phytochemicals (i.e., polyphenols, carotenoids, antocyans, vitamins, etc.) with great UV-filtering activity, as alternative for the synthetic counterparts. These compounds act against the ROS, being able to provide broad-spectrum products with anti-UV, antioxidant, anti-inflammatory and wound-healing properties [10].

Herbal extracts have expanded their area of applicability from highly fluorescence extract-based nanomaterials [19,20] towards vegetable lipid nanoparticles with antioxidant and antitumor activities [21,22,23]. The use of active phytoconstituents from herbal extracts is beneficial in combating the deleterious effects of UV rays. Herbal extracts produce healing, softening, rejuvenating, antiaging, antioxidant and sunscreen effects [24,25]. This approach in association with the lipid nanotechnology represents the latest research in the field of photoprotection corelated with the chemistry of the herbal extracts. The main features of applied lipid nanotechnologies in cosmetic active ingredient delivery area include specific targeting, decreasing toxicity while assure beneficial effects and more biocompatibility [26,27]. Besides these, the nanosized carriers applied on the skin provide an occlusive effect which leads to an increase in skin hydration and can promote the deposition of active ingredients into the viable skin [28].

Considering the points stated above, the current investigation aimed to design and synthesize stable and functional bioactive lipid nanocarriers (BLN)-based hydrogels for UVA and UVB radiations absorption, targeting high-level photoprotection and a high degree of skin hydration, along with low-level toxicity of the final products. Of particular interest is the question of how the presence of herbal extracts can influence the photoprotective and hydration properties, to achieve an enhanced therapeutic effect. To decrease the concentrations of sunscreens incorporated in the formulations as well as to identify some formulas with high skin efficiency, along with the two UVA and UVB sunscreens, butyl-methoxydibenzoylmethane (BMDBM) and 2-ethylhexyl-2-cyano-3,3-difenil acrylate or octocrylene (OCT), two herbal extracts, *willow bark extract* (*Wbe*) and *ivy leaves extract* (*Ile*) were co-loaded in lipid nanocarriers based on health-promoting *carrot oil (Co*) and *pomegranate oil (Po).* BLN prepared were suitably characterized for particle size, zeta potential, thermal behavior, and entrapment efficiency. *In vitro* UVA and UVB filters release, rheology and viscosity studies were carried out. Furthermore, the effect of BLN-*Ile*/*Wbe*-UV-filters based hydrogel upon the skin hydration level and photoprotective action was investigated, obtaining insight into its potential related to the antiaging effect. Thus, the cosmetic efficiency of the BLN-*Ile* and BLN-*Wbe* hydrogels loaded with UV-filters was investigated, determining the *in vitro* sun protection factors, for both UVB and UVA radiations (SPF and UVA-PF). Moreover, *in vitro* photostability tests were conducted for the studied formulations, after a controlled irradiation stage.

## 2. Materials and Methods

### 2.1. Materials

Polyoxyethylenesorbitan monolaurate (Tween 20) was purchased from Merck (Darmstadt Germany), while Poloxamer 188 and L-α-phosphatidylcholine were bought from Sigma Aldrich Chemie GmbH (Munich, Germany). Glycerol monostearate (*GMS*) and cetyl palmitate (*CP*) were purchased from Cognis GmbH (Monheim am Rhein, Germany) and Acros Organics (New Jersey, NJ, USA), respectively. The herbal extracts—*Ivy leaves extract*, *Ile* (14.65% Hederacoside C), and *Willow bark extract, Wbe* (10% Salicin)—were supplied by S.C. Biopharm, Bucharest, Romania. The two herbal extracts (supplied by S.C. Biopharm, Bucharest, Romania) are standardized extracts in the main active ingredient: *Ivy leaves extract* (*Ile*) is standardized based on the content of *Hederacoside C*, e.g., 14.65% Hederacoside C; *willow bark extract* (*Wbe*) is standardized based on the content of Salicin, e.g., 10% Salicin. The two vegetable oils—*Carrot oil* (*Co*) and *Pomegranate oil* (*Po*)—were provided by the Romanian Herbal Company, Hofigal S.A (Bucharest, Romania). The fatty acids composition was determined by gas chromatography, using the derivatization method (Agilent Technologies model 7890A instrument (Santa Clara, CA, USA), coupled with an Agilent Technologies model 5975 C VL MSD (Santa Clara, CA, USA), mass detector with Triple Axis Detector and Agilent auto-sampler (Santa Clara, CA, USA); the separation into components was made on a capillary column Supelco SP^TM^ 2560 (Sigma Aldrich Chemie GmbH, (Munich, Germany)): 6.15% linoleic acid (ω-6), 5.05.% oleic acid (ω-9), 2.87% palmitic acid, 1.99% stearic acid, 82.85% punicic acid (ω-5) for pomegranate oil and 65.07% linoleic acid (ω-6), 25.67% oleic acid (ω-9), 5.53% palmitic acid, 3.22% stearic acid for carrot oil. For the hydrogel formulations, Carbopol 940 (99% purity), triethanolamine (98% purity) and glycerol were supplied by Sigma Aldrich Chemie GmbH (Munich, Germany).

### 2.2. Methods

#### 2.2.1. Preparation of Bioactive-Loaded Nanostructured Lipid Carrier

The optimized free and loaded BLN were prepared using as lipid matrix mixture of cetyl palmitate (CP), glycerol monostearate (GMS) and vegetable oils (*carrot oil*, *Co*, and *pomegranate oil*, *Po*), by high pressure homogenization (HPH) strategy, following the procedure previously described by our group [29,30]. Briefly, an aqueous phase consisting of a mixture of surfactants (Tween 20, Phosphatidylcholine and Synperonic PE/F68) and herbal extract (*Ile* and *Wbe*) was added to the melted lipid phase (mixture of GMS, CP, *Co* and *Po*, at 75 °C), containing also the UV-filters (BMDBM and OCT). Total lipid phase concentration, consisting of solid fat and liquid oil, was kept constant at 10%. The composition of BLN formulations is presented in Table 1.

After 15 min of stirring at 75 °C, the resulting emulsions were subjected to a high shear homogenization stage (High-Shear Homogenizer PRO250 type, Monroe, CT, USA) by applying 12,000 rpm for 1 min and then processed through HPH (APV 2000 Lab Homogenizer, (Charlotte, NC, USA), with six homogenization cycles (196 s) at 500 bars.

The resulting hot nanoemulsions were cooled to room temperature to allow recrystallization of the lipid phase, resulting in bioactive lipid nanocarriers co-loaded with plant extract and UV filters. The aqueous dispersions of BLN were frozen at –25 °C overnight and then underwent a lyophilization step (0.05 mbar, −54 °C, 72 h) using an Alpha 1–2 LD Freeze Drying System (Osterode am Harz, Germany), with obtaining of solid formulations of BLN.

#### 2.2.2. Particle Size, Morphology, and Zeta Potential Analysis

The analysis of the mean particle diameter (Zave) and the size distribution (PdI) of BLN formulations was performed by photon correlation spectroscopy (PCS), using a Zetasizer ZS 90 (Malvern Instruments Inc., Worcestershire, UK), equipped with a solid-state laser (670 nm) at a scattering angle of 90° and a temperature of 25.0 ± 0.1 °C. Samples were prepared by dispersing the BLN-*Ile*/*Wbe*-UV-filters in appropriate amount of deionized water before conducting the experiment. The particle size data were evaluated using intensity distribution. Each value of *Zave* and *PdI* was given as average of three individual measurements.

The morphological aspects of the BLNs co-loaded with herbal extract and UV-filters have been observed by transmission electron microscopy (TEM, Hitachi High-Tech Corporation, Tokyo, Japan). To perform TEM analysis, a Hitachi HD 2700 Scanning Transmission Electron Microscope (Hitachi High-Tech Corporation, Tokyo, Japan) was used; for appropriate analysis, aqueous dispersions of BLN were diluted in distillate water and deposited on standard Cu grids with Carbon thin layer film. TEM images were obtained in bright field mode.

The electrical charge (zeta potential, *ξ*) of lipid nanocarriers was determined in a capillary cell, using the same instrument which utilized the Helmholtz–Smoluchowski equation to convert the measured particle electrophoretic mobility into zeta potential. Prior to analysis, BLN were diluted 1:100 with deionized water and adjusted with 0.9% NaCl solution, to avoid multiple scattering effects and to reach a conductivity of 50 µS/cm, respectively. *ξ* was computed using Helmoltz–Smoluchowski equation:(1)$$\xi =EM\;\frac{4\pi \eta }{\varepsilon }$$
where, *ξ* is the zeta potential, *EM* is the electrophoretic mobility, *η* is the viscosity of the dispersion medium, and *ε* is the dielectric constant.

#### 2.2.3. Differential Scanning Calorimetry Analysis

The thermal behavior of lipid nanocarriers was studied by differential scanning calorimetry (DSC), using a differential scanning calorimeter Jupiter, STA 449C (Netzsch, Selb, Germany). The samples (≈10 mg) were weighed directly into standard aluminum pans and heated between 20 °C to 100 °C with a scan rate of 5 °C/min. An empty aluminum pan was used as reference. The onset temperature, melting point (peak maximum), and melting enthalpy (ΔH) were calculated using the Proteus software, provided by Netzsch (NETZSCH Proteus^®^ software V4.8.2, Selb, Germany).

#### 2.2.4. Entrapment Efficiency and Drug Loading

The entrapment efficiency (EE%) of the UV-filters within the BLN was determined by quantifying the actives encapsulated into bioactive lipid nanocarriers versus the amount of the free/un-entrapped OCT and BMDBM present in the ethanolic medium, employing the UV–Vis spectrometry. Each BLNs sample (0.5 g) was dispersed into ethanol (1 mL) and the resulting suspension was centrifuged for 25 min at 13,000 rpm, (Sigma 2K15, Osterode am Harz, Germany). The supernatant (containing the un-entrapped OCT and BMDBM) was collected and diluted with ethanol. The obtained solution was evaluated at λ = 303 nm (for OCT) and λ = 356.5 nm (for BMDBM), using an UV-Vis-NIR Spectrophotometer type V670 (Jasco, Tokyo, Japan).

The efficiency of BLN to entrap *willow bark extract* and *ivy leaves extract* has been achieved by HPLC method. *Willow bark extract* entrapment efficiency (*EE*) was determined by quantifying the *Salicin* encapsulated into BLN, while EE of *ivy leaves extract* was evaluated by quantitative determination of *Hederacozide C* from BLN. The lyophilized-BLNs have been processed as previous described for UV-Vis determination for obtaining a supernatant which contain un-entrapped *Ile*/*Wbe*. The collected supernatant was analyzed using a Jasco 2000 liquid chromatograph equipped with a Nucleosil C18 column (25 × 0.4 mm) and a UV detector at λ = 205 nm. The mobile phase was composed by ACN: H_3_PO_4_ 0.5 % (70:30), the retention time of *Salicin* and *Hederacoside C* was 1.7 min and 12.6 min, respectively, and the flow rate was 1 mL/min.

The entrapment efficiency (EE%) was calculated according to the following equation:(2)$$\mathrm{EE}\%=\frac{W_{a}-W_{s}}{W_{s}}\times 100$$
where *W_a_* represents the weight of UVA/UVB filters and *Ile*/*Wbe* added into nanocarriers, and *W_s_* is the analyzed weight of UV-filters/herbal extract in supernatant.

The drug loading (%DL) was determined using the equation:(3)$$\%\;\mathrm{DL}=\frac{W_{a}-W_{s}}{W_{a}-W_{s}+W_{L}}\;\times \;100$$
where *W_a_* is the weight of the active principle (OCT, BMDBM, *Ile* or *Wbe*) added in the nanocarriers, *W_s_* is the analyzed weight of active in supernatant, and *W_L_* is the weight of lipids added in the nanocarriers.

#### 2.2.5. Preparation of Loading Hydrogels with BLN-Wbe/Ile-UV-Filters

Considering the future determinations of the photoprotective action, the release behavior of the two sunscreens and the hydration effect, the bio-nanocarriers were conditioned in the form of topical hydroalcoholic formulations, using Carbopol hydrogel: BLN ratio = 2:1). Conditioning was performed in two stages, as follows: (i) preparation of the support-gel by gradually adding of Carbopol 940 (1 g) to an aqueous solution containing glycerol (12 g), ethanol (5 g) and water (70 g), under stirring; the solution was allowed to rest for 24 h and then the pH was adjusted at 6, using triethanolamine solution (10% wt); finally, 30 g of water was added to complete the formulation; (ii) incorporation of the lyophilized BLNs into the hydrogel by gradually adding of small amounts of BLN-*Wbe/Ile*-UV-filters (in a mass ratio of 1:2), at 500 rpm. The final cosmetic formulations contain 3.2% OCT, 2.2% BMDBM, 3.7% herbal extract and 13.8% *Co*/*Po*.

#### 2.2.6. In Vitro Release Experiments

The release rates of the UVA and UVB filters from the prepared BLN-based hydrogels were determined through Franz diffusion cells (25 mm in diameter; Hanson Research Corporation, Chatsworth, CA, USA), using cellulose membranes (0.1 µm; Whatman, Germany). This technique is known as a suitable method to assess drug release from topical formulations. Franz diffusion cell consists of a donor and a receptor chamber between which a cellulose membrane (0.1 µm; Whatman, Germany) is placed. The receiving solution consisted of ethanol buffer solution (phosphate-buffered saline, pH 5.5/ ethanol, 50:50, *v*/*v*) for ensuring pseudo-sink conditions by increasing sunscreen solubility in the receiving phase. The cellulose membranes were moistened by immersion in ethanol buffer solution for 1 h at room temperature before being mounted in Franz-type diffusion cells. The area of diffusion/surface area was 0.636 cm^2^ and the receiving chamber volume was 6 mL. The receptor phase was constantly stirred at 400 rpm and thermostated at 37 °C to maintain the membrane surface at 32 °C, to mimic skin temperature.

Each formulation (250 mg) was applied on the membrane surface and the experiments were run for 24 h. At different time intervals (1, 2, 3, 4, 5, 6, 7, 8 and 24 h), samples of the receiving solution (500 µL) were withdrawn and replaced with an equal volume of receiving solution pre-thermostated at 37 °C. Samples of the receptor phase were analyzed for UVA and UVB filters concentration by using UV spectrophotometry at λ_max_ 356.5 nm and 303 nm, respectively. The release kinetics of UV filters were depicted using the following mathematical models equations: first order: $\log\left(100-\%R\right)=k_{1}t/2.303$; Higuchi: $\%R=k_{2}\sqrt[{2}]{t}$; Hixson–Crowell: $\sqrt[{3}]{100-\%R}=k_{3}t$, where *%R* is percent of UV filter release at time *t, k_1_*, *k_2_* and *k_3_* are the rate constants for first order, Higuchi and Hixson-Crowell, respectively, and *n* is the release exponent. The mathematical models’ equations with the biggest R^2^ were selected as the best release kinetic model.

#### 2.2.7. In Vitro Permeation Test

The *in vitro* permeation experiments were performed in the Franz diffusion cells systems employing a collagen skin membrane as a skin model. The collagen membrane (received from National Institute of Research and Development for Biological Sciences, Bucharest, Romania) was rehydrated by immersion in distilled water at room temperature for 1 h before being inserted in Franz diffusion cells. The receptor compartment was filled with buffer solution/ethanol (50/50 *v*/*v*) to establish sink conditions and to assure a permeant UV-filters solubilization; the receptor solution was stirred at 500 rpm and thermostated at 32 °C during all the *in vitro* tests. Then, 20 mg BLN-based hydrogels were placed on the collagen membrane in the donor compartment. At predetermined time intervals, 500 mL of receiving solution was withdrawn and replaced with fresh solution. Each BLN-hydrogel was run in triplicate for 24 h, using three different donor compartments, and the results were expressed as the cumulative amount of drug permeated (µg/cm^2^) ± SD. The samples were analyzed for UVA and UVB filters content by UV-Vis spectroscopy as described in Section 2.2.4. The skin flux of each UV-filter was calculated by plotting the cumulative amounts of UV-filter penetrating the skin against time and determining the slope of the linear portion of the curve. The UV-filters fluxes (mg/cm^2^/h), at steady state, were calculated by dividing the slope of the linear portion of the curve by the area of the skin surface through which diffusion took place.

#### 2.2.8. The In Vitro SPF and UVA-PF Determinations

The *in vitro* determination of the sun protection factor (SPF) and of the erythemal UVA protection factor (UVA-PF) has been achieved on different BLN-cosmetic hydrogels (containing 2.26% BMDBM and 3.20% OCT), using a UV–Vis V670 Spectrophotometer Jasco (Jasco, Tokyo, Japan) equipped with integrated sphere and adequate soft, based to Diffrey and Robson method [31]. The hydrogel formulation based on BLN-*Wbe/Ile*-UV-filters (2 mg/cm^2^) was applied by a Transpore^TM^ membrane that imitates the natural skin. The UV absorption spectrum of each sample was registered from 290 to 400 nm, by using as a reference support the Transpore membrane tape without BLN. To calculate the SPF and UVA-PF values, the following equations have been used:(4)$$\mathrm{SPF}=\frac{\sum_{290}^{400}E_{\lambda }B_{\lambda }}{\sum_{290}^{400}E_{\lambda }B_{\lambda }/MPF_{\lambda }}$$
(5)$$\mathrm{UVA}-\mathrm{PF}=\frac{\sum_{320}^{400}E_{\lambda }B_{\lambda }}{\sum_{320}^{400}E_{\lambda }B_{\lambda }/MPF_{\lambda }}$$
where *E_λ_* represents the spectral irradiance of terrestrial sun light under defined conditions, *B_λ_* is the relative erythemal effectiveness, *MPF_λ_* is the monochromatic protection factor for selected wavelength (the difference between the spectrum of measured sample applied on support and support spectrum).

#### 2.2.9. Photostability Studies

Photostability studies were performed on each developed BLN-based hydrogel formulation containing BMDBM (2.26.%, *w*/*w*) and OCT (3.20%, *w*/*w*). In order to investigate the SPF and UVA–PF, the selected samples were irradiated using a BioSun system (Vilver Lourmat, France), as follows: at 365 nm, 3 h (UVA), at 312 nm, 5 h (UVB). The irradiation energy was the 19.5 J/ cm^2^ for both tested domains.

#### 2.2.10. Rheological Behavior

To assess the rheologic behavior of the compositions, dynamic oscillatory measurements were performed on a Kinexus Pro rheometer (Malvern, Worcestershire, UK) equipped with Peltier element, at a pre-established temperature of 32 °C. A cone-plate geometry was used with an upper plate radius of 40 mm and an angle of 4°, at a fixed gap size of 0.15 mm. Dehydration was prevented by using a water-lock. The storage (G′) and loss (G″) moduli were plotted in logarithmic scale. In a first step, amplitude sweep tests were performed in order to establish the linear viscous region (LVR) of the compositions. In this respect, the samples were subjected to an increasing oscillatory stress (10^−1^ ÷ 10^2^ Pa) while temperature and frequency were kept constant (32 °C, 1 Hz). Subsequently, frequency sweep tests were performed keeping the oscillatory deformation constant, at a lower value then the LVR limit. The frequency was gradually decreased from 10 to 0.10 Hz.

#### 2.2.11. In Vivo Determination of the Hydration Degree

The hydration effect of the developed BLN-hydrogels on the upper layers of the skin was assessed by using a Multi Skin Test Center MC 1000 (Köln, Germany). The assay was performed on different skin regions of human volunteers (arms, legs, hands).

Prior to BLN-hydrogel application, the skin was well cleaned, and the hydration skin degree was measured using a moisture sensor. Then, BLN-hydrogels were applied at 4 h for 28 days, monitoring the hydration degree for the skin.

The pre-clinical examination after the human volunteers used the BLN-hydrogel developed within the study was performed according to the protocol and internal procedures of a national pre-clinical testing center. The study and the pre-clinical data results were carried out in accordance with the principles of good clinical practice (International Recommendations ICH E6 (R1) of 10/06/1996, Directive of the European Parliament 2001/20/EC-OJ/EC from 01 May 2001) and the Declaration from Helsinki (June 1964). Additionally, the Colipa recommendations “Guidelines for the evaluation of the efficacy of cosmetic products, May 2008” and the Regulation of the European Parliament and of the Council (EC) no 1223/2009 of 30 November 2009 on cosmetic products have been respected.

#### 2.2.12. Statistical Analysis

The obtained data for the current research study were expressed as the mean value of three individually measurements ± standard deviation (SD). The statistical significance of the experimental data was determined using the ANOVA test.

## 3. Results

### 3.1. Size and Physical Stability Features

The size and morphology characterization of the lipid bio-nanocarriers revealed the obtaining of delivery systems with average diameters between 73 and 171 nm (Figure 1 and Table 2). The use of *carrot oil* for the preparation of BLN led to smaller average diameters than for *pomegranate oil* (e.g., 73.5 nm ± 0.72 and *PdI* = 0.138 for BLN*1*/prepared with *carrot oil* versus 143.4 nm ± 1.82 and *PdI* = 0.232 for BLN*2*/made with *pomegranate oil*), most likely the high content of punicic acid (83% ω-5) in *pomegranate oil* being responsible for increasing the size of lipid core. The combination of *carrot oil* and the mixture of solid lipids also led to a narrow distribution of the lipid particle population, most BLN formulations having polydispersity index values less than 0.14. After encapsulation of the three active principles, the mean diameters increased significantly (e.g., *Zave* = 120.2 ± 0.76 nm, *PdI* = 0.138 ± 0.01 for BLN*1-Ile*-UV-filters and *Zave* = 171.3 ± 0.76 nm, *PdI* = 0.203 ± 0.008 for BLN*2-Ile*-UV-filters). The accommodation of *Wbe* together with UV-filters resulted in smaller size than in the case of previous BLN-*Ile*-UV-filters, predictable results considering the complex structure of the main polyhydroxy component in the *ivy extract*—Hederacozide C—which requires larger holes for accommodation in the lipid core. Although no studies have been reported in the literature on the co-optation of herbal extracts together with OCT and BMDBM filters in the same nanostructured distribution system, in related research aimed at encapsulating BMDBM [32] or BMDBM and OMC [33] into lipid nanocarriers, led to comparable values of particle size, e.g., smaller than 150 nm.

Regarding the physical stability of the bioactive lipid nanocarriers, the combination of both vegetable oils with the solid lipid mixture led to variable electronegative zeta potentials; for instance, for the free BLN, the zeta values were −32.9 ± 0.50 mV for BLN*1* and −34.8 ± 0.31 mV for BLN*2*. The advanced development of negative charges on the lipid particles surface has been shown for lipid bio-nanocarriers that co-encapsulate the three herbal and synthetic active ingredients, the zeta potential values decreasing significantly compared to free BLNs (Table 2). It is interesting to note that as compared to literature data, the ξ-potential values of BLN-*Wbe/Ile*-UV filters are more electronegative (e.g., −51.2 ± 1.56 mV, for BLN*2*-*Wbe*-UV-filters and −39.2 ± 0.61 mV for BLN*1*-*Wbe*-UV filters) than lipid nanocarriers loaded with BMDBM or co-loaded with BMDBM and OMC, for which values of −20.9 mV and −28.8 mV have been determined [32,33]. These results suggest that the selected vegetable oils (Pso and Co) and surfactants strongly influences the charges distribution on the lipid particles surface leading to a pronounced electronegative double layer and thus better stability.

### 3.2. Physico-Chemical Characterization of BLN Loaded with the Herbal Extract and UV-Filters

The structural rearrangement of the lipid core after the encapsulation of the herbal extract and of the two sunscreens is evidenced by comparing the DSC behavior of the free-BLNs and those that co-encapsulate both types of active principles. Melting point decrease and a change in the corresponding enthalpy for the BLN-*Ile*/*Wbe*-UV-filters were detected by decreasing the endothermic profiles (Figure 2). The presence of *Wbe/Ile* and the two sunscreens (OCT and BMDBM) caused a change in the melting enthalpy, e.g., from 98.1 J/g (for BLN*2*) to 116.8/118.6 J/g (for BLN*2*-*Wbe/Ile*-UV filters), together with a decrease in the melting point (e.g., from 49.4 °C for BLN*2* to 47.9 °C in the case of BLN*2*-*Wbe/Ile*-UV filters), which suggests a lower organization level in the amorphous BLN network that co-encapsulates the two natural and synthetic actives. This behavior indicates that BLN synthesized with the two vegetable oils is characterized by a lipid network with many imperfections and, consequently, with a significant effect on the encapsulation efficiency of the sunscreens.

Noticeable is the significant disturbance of the lipid core consisting of *carrot oil*, cetyl palmitate and glycerol monostearate; the drastic decrease in the endothermic peak encountered in BLN*1* justifies the existence of a complex reorganization of the lipid core after the encapsulation of *Wbe/Ile* and the sunscreens. This last statement is also supported by the appearance of the second melting shoulder, located at approx. 53 °C.

Quantitative determinations revealed a higher BMDBM encapsulation efficiency in all BLNs prepared with 30% *Co/Po* (relative to total lipid concentration) compared with OCT, ranging from 76.4% to 89.5% BMDBM and between 57.8 and 84.4% OCT (Table 2). The best encapsulation capacity of the two sunscreens was found in the case of NLC*2-Wbe*-UV-filters, where 89.50 ± 1.18% BMDBM and 84.4 ± 0.45% OCT were determined.

In the case of capturing the two herbal extracts, the encapsulation efficiency values, between 80.1 and 82.2% for Hederacoside C and 89.2 and 90.8% (Table 2), for Salicin demonstrate an efficient adaptability of *Ile*/*Wbe* in the lipid nanocarriers based on *carrot* and *pomegranate oils*. Most likely, the hydrophilic nature of *Ile*/*Wbe* allowed the retention of a significant amount of herbal extracts in the shell surfactants, in addition to accommodating *Ile*/*Wbe* together with OCT and BMDBM inside the lipid matrix. As was predicted, the complex structure of the Hederacozide C from *Ile*, with multiple hydroxy groups, resulted in a slight decrease in EE, for example EE for *Ile* from BLN*2*-*Ile*-UV-filters was 82.3% ± 2.20, compared to EE for *Wbe:* 90.8 ± 1.18%, from BLN*2*-*Wbe*-UV-filters.

The main factors that could be responsible for the encapsulation efficiency could be related to the chemical structure of the encapsulated active molecules (OCT/BMDBM/ *Ile*/*Wbe*) with specific distribution within the lipid core and/or surfactant shell and also to the vegetable oils composition, with various unsaturated fatty acids. For example, the high amount of ω-5 (~83% punicic acid) and a low ω-6 and ω-9 fatty acids content (e.g., 5% ω-9 and 6% ω-6) characteristic for *pomegranate oil* provides some imperfections within the lipid network, more suitable to catch the active molecules as compared with the *carrot oil*, free of punicic acid but with high content of ω-6 (65% linoleic acid).

The loading capacity of the BLNs varied from 3.8 to 4.2% for UV-A filter (BMDBM) and from 3.9 to 5.6% for UV-B filter (OCT), respectively (Table 2), confirming the great capacity of the lipid matrix to encapsulate BMDBM than OCT. The results obtained for herbal extracts, indicate their lower accommodation level within the lipids arrangement. This result is due to the hydrophilic affinity of *Wbe/Ile* that prefer to remain outside of the lipid core.

### 3.3. Comparative Evaluation of the Release Profiles of the Two UV-Filters

A first appreciation of the *in vitro* release profiles of OCT and BMDBM from developed topical BLN-formulations/hydrogels, is the observation of a much slower degree of BMDBM release compared with OCT (Figure 3). The slow release of BMDBM from the BLN-based hydrogel suggests its homogeneous capture in the lipid core, previous demonstrated by the higher values of the BMDBM encapsulation efficiency, compared to OCT.

BLN-based hydrogels co-encapsulating *ivy extract* and sunscreens prevent the occurrence of a sudden release of OCT or BMDBM, both UV filters being released in a controlled manner, with an almost constant degree for OCT/BMDBM release (Figure 3a,b).

Regarding the influence of vegetable oil, the amount of sunscreen released was significantly higher from BLN prepared with carrot oil (e.g., 20.3% OCT from BLN*1*, compared to 14.8% OCT from BLN*2*, after 8 h of experiments). The same behavior is observed in the case of BMDBM release (e.g., 8.2% BMDBM from BLN*1* versus 5.6% BMDBM from BLN*2*). The influence of vegetable oil type on the release rate of OCT and BMDBM from lipid nanocarriers prepared with *rice bran oil* and *raspberry seed oil* was also highlighted by Niculae et al. [34]. The authors demonstrated that OCT release (e.g., up to 17% OCT release) and BMDBM release (e.g., up to 3.9% BMDBM release) was amplified by the presence of *rice bran oil*. Although the *in vitro* release behavior follows a comparable trend, the presence in our study of herbal extract/*ivy extract* co-opted with UV filters led to a faster release of OCT (e.g., up to 25% OCT for BLN*2-Ile*-UV-filters) and BMDBM (e.g., up to 10% BMDBM for BLN*1-Ile*-UV-filters), compared to previous results obtained by Niculae et al. These differences can be attributed to a faster degradation of the surfactant coating that hosts the herbal extract, which facilitates the faster exit of the UVA and UVB filter from the lipid core. The slightly faster *in vitro* release profile encountered for BLN*1* can be also attributed to the smaller particle size. For small particles, the saturation solubility increases significantly. Both, the increase in saturation solubility and the widening of the surface area contribute to the increase in the dissolution rate and, consequently, the speed of release would be expected to be faster.

It can also be seen from Figure 3c,d that HG based on BLN-*Wbe*-UV-filters showed a significant slow-release profile, high concentrations of OCT and BMDBM being maintained inside the nanocarrier systems. For instance, the HG-BLN*2-Wbe*-UV-filters had the lowest release rate, 9.1% OCT/8 h and 10.4% OCT/24 h, respectively, 4.2% BMDBM/8 h and 5% BMDBM/24 h; moreover, this slow release is also supported by the highest values of the encapsulation efficiencies (84% OCT and 89% BMDBM) and most likely by the existence of a structural model of BLN, of type “enriched core in drug”, which promotes a slow release. The release of minimal amounts of sunscreen is beneficial due to the preservation of the UV filters in the formulations, with a low potential to pass through the skin and reach the systemic circulation.

By evaluation of kinetic release parameters of OCT and BMDBM from hydrogels based BLN-*Ile*/*Wbe*-UV-filters (e.g., release rate constant, *k*; the release exponent, *n*, as well as the correlation coefficient, *R^2^*) it has been shown that the BLN-*Wbe/Ile*-UV-filters assures a kinetic release which correspond to the Higuchi model (for instance, in case of OCT, R^2^ = 0.9183 and k_2_ = 4.5780 h^−1^ for BLN*1*-*Wbe*-UV-filters and 0.9018, k_2_ = 4.9239 h^−1^ for BLN*2*-*Ile*-UV-filters; the same behavior has been encountered for BMDBM: R^2^ = 0.9254, k_2_ = 2.0090 h^−1^ for BLN*1*-*Wbe*-UV-filters and R^2^ = 0.9009, k_2_ = 2.5848 h^−1^ for BLN*1*-*Ile*-UV-filters). The Higuchi model describes a diffusion process governed by Fick’s law.

### 3.4. In Vitro Permeation Study

The efficacy of UV-filters depends on their ability to remain on the skin surface after topical application of a sunscreen product, with or without minimum permeating into the skin layers. Figure 4 shows the different profiles for the BLN-hydrogels, in terms of the cumulative amount of permeated OCT and BMDBM, after 24 h. Significant differences were observed between the behavior of the two categories of UVA and UVB filters. A higher amount of penetrated OCT into the receptor phase of Franz diffusion cells was detected for hydrogel formulations such as BLN-*Ile*-UV-filters, i.e., a difference almost double that between the cumulative amount of OCT and BMDBM in the BLN*2*-*Ile* formulation (Figure 4a). The more lipophilic character of OCT (logP = 6.9) than BMDBM (logP = 4.5) could be a reason for the enhanced permeability of OCT compared to BMDBM [35]. In addition, the higher “substantiality” of BMDBM to form weak hydrogen interactions (due to the two co-planar carbonyl groups) with the core-lipid matrix or hydroxyl groups in *Ile*, could be responsible for the minimizing of BMDBM percutaneous absorption. The different affinity of UV-filters for skin model membrane may also explain the different behavior obtained for OCT and BMDBM. In two related studies, it has been reported that lipid nanoparticles have dramatically reduced the permeation of UV filters through the skin and favored their localization in the superficial layers of the skin [12,33].

A distinctive interest is the analysis of flux values, at the steady state, of OCT and BMDBM in the case of formulations prepared by associating UV filters with the two phytochemicals—*Wbe* and *Ile* (Figure 4b). When UV-filters are incorporated into BLN-*Wbe*, a lower flux has been determined, especially for BMDBM. The low flux value for BMDBM at the steady state was also previous obtained by Puglia et al. when formulating a blend of two filters into nanostructured lipid carriers [33]. Among the tested UV-filters blends, BMDBM reached a reduced flux value of 0.71. By comparing our flux values for BMDBM (e.g., between 0.25 and 1.12) with that reported by Puglia et al., a potential influence of the herbal extract-*Ile*/*Wbe* can be assumed (Figure 4b). For example, in case of BLN*2-Wbe*-UV-filters based on *pomegranate oil*, the lower flux of 0.25 could be correlated with previous quantitative results (Table 2), where better *Wbe* capture efficiencies were determined than in the case of BLN-*Ile*-UV-filters.

By comparing the flow values obtained from BLN*1* and BLN*2* in terms of the influence of the type of *carrot* or *pomegranate oil*, with few exceptions, they did not produce any appreciable increase in the amount of UV-filters penetrating through skin membrane model (Figure 4b). However, the differences become significant when comparing OCT flow values with those of BMDBM, especially in BLN-hydrogels associated with *willow bark extract.* When OCT and BMDBM are incorporated into BLN-UV-filters-*Wbe* the skin permeability of the UV-filters has been drastically reduced, which emphasizes a delay in skin permeability, UV-filters remaining mainly on the skin surface. The marked decrease in the flow for BMDBM and OCT from BLN*2*-*Wbe* compared to BLN*1*-*Wbe*, obtained by dividing the value of the flow of OCT/BMDBM from BLN*1*-Wbe (containing *carrot oil*) compared to that obtained from BLN*2*-*Wbe* (containing *pomegranate oil*), turned out to be 1.75 for OCT, while the flow decrease for BMDBM was up to 3.6. These results are in line with assumptions in the references, namely, the different plant extracts incorporated in topical nanoformulations lead to improved SPF and antioxidant activity [36].

### 3.5. In Vitro Determination of the Blocking Effect of UV-A and UV-B Radiation of the Developed Topical Formulations

The developed BLN-based hydrogels which contain minimal amounts of sunscreens, e.g., 3.20% OCT and 2.26% BMDBM, and significant amount of vegetable oils, 7% *carrot oil/pomegranate oil*, manifest important anti-UV properties. Many studies reported that the presence of natural compounds from herbals with lipid nanoparticles may offer a synergistic effect in terms of UV absorbance and antioxidant capacities [21,37]. The SPF and UVA-PF values determined for two of the four developed hydrogel formulations demonstrate the benefits of using these mixed systems of *carrot oil* and *ivy extract*/*willow extract* in combination with minimal amounts of synthetic sunscreens (Figure 5). The SPF values obtained for the two hydrogels with BLN*1-Ile/Wbe*-UV-filters of 13 and 12.1, respectively, provide average protection against UVB radiation (with 90% absorption of UVB radiation). In a related study, Nikolic et al. formulated gel containing carnauba wax-based nanostructured lipid carriers (NLC) with a mixture of three organic UV filters (bis-ethylhexyloxyphenol methoxyphenyl triazine, ethylhexyl methoxycinnamate, and ethylhexyl triazone). Their NLC-formulations exhibited SPF values of 20.1 and 14.1 [38]. Compared to these results, it is noteworthy that the lipid nanocarriers-based hydrogels developed in our research reveal a significant increase in the anti-UV effect after a controlled irradiation (Figure 5). For instance, a twice more effective photoprotection was determined for BLN*2*-*Wbe/Ile*-UV-filters prepared with *carrot oil* (e.g., SPF after irradiation = 24, which revealed a 97% absorption of UVB radiation). These results could be associated with the *carrot oil* which allows an increase in the effectiveness of sun protection, by ensuring an efficient distribution and release of the two UV-filters from the lipid matrix and by their own anti-UV properties. The beneficial effect of vegetable oils was also mentioned by Niculae et al., *rice bran oil- and raspberry seed oil*-based lipid nanocarriers resulted in a more efficient exercise of the photoprotective than those based on the solid–lipids blend [34].

In the UVA domain, the sun protection of hydrogels based BLN1*-Ile/Wbe*-UV-filters is more pronounced as compared with the UVB domain, the UVA-PF reaching values of 30.7, respectively 30, which denotes an absorption of 98% UVA radiations.

For hydrogels with BLN*2* (prepared with *pomegranate oil*), despite the SPF values being similar to those of the BLN1 (prepared with *carrot oil*), the protection against UVA radiation (due mainly to the presence of BMDBM) decreases significantly for the two types of BLN*2* (Figure 5). There are two aspects that can be considered for the drastic variation of UVA-PF in the case of BLNs prepared with the two natural oils:(i)Different distribution of BMDBM in the lipid network (as reported in the case of *in vitro* release data). According to the results obtained in the quantitative determinations and the release study, in the hydrogel formulation, BMDBM is better captured in the lipid core of BLNs (higher encapsulation efficiencies, compared to those obtained in the case of OCT), and as such, it is harder to get out to manifest the photoprotective effect (slower release of BMDBM, according to the release data).(ii)Existence of a BMDBM distortion (known as a high sensitivity UV-A filter). High ω-6 fatty acid content (65% linoleic acid) in *carrot oil* can prevent BMDBM distortion, compared to *pomegranate oil* (with low ω-6 fatty acid content, 6% linoleic acid).

The photoprotective behavior of hydrogels containing BLNs subjected to irradiation underwent modifications as follows: *(i*) UVB protection increased significantly after irradiation for all the formulated hydrogel (e.g., from initial SPF values of 12 and 14, SPF have reached of about 20 for hydrogels containing BLN*2*-*Wbe/Ile*-UV-filters and ~24, in the case of BLN*1*-*Wbe/Ile*-UV-filters; *(ii*) for hydrogels containing BLNs prepared with *Co*, a slight decrease in UVA-PF was observed, compared with the initial value (UVA-PF_initial_ = 30 versus UVA-PF_after irradiation_ = 24), whereas the BLN systems prepared with *Po*, after irradiation they showed a significantly higher anti-UVA effect (from UVA-PF_initial_ between 5 and 7 to UVA-PF_after irradiation_ between 18 and 20).

### 3.6. Rheological Behavior and Hydration Effect of Topical Formulations Based on BLN-Wbe/UV-Filters

The attribution of the rheological behavior of the BLN-*Wbe/Ile*-UV-filters based hydrogels was performed by dynamic oscillatory measurements. Firstly, amplitude sweep measurements were performed. As described in [39], the registered behavior can be correlated with the firmness of the compositions. The data indicated that all composition show a firm structure, with a linear viscous region ranging over more than two decades (Figure 6). In agreement with the results presented in [40], the incorporation of fillers leads to an increased value of both G′ and G″ in BLN*1-Wbe*-UV-filters and BLN*2-Wbe*-UV-filters when compared to the control sample. Furthermore, these results indicated that BLN*2-Wbe*-UV-filters formulation is the most structured composition, being able to resist to the applied stress. When performing stress sweeps at low stress values, resistance to stress may be correlated to the formulations spreadability; our results indicated that the addition of BLN*-Wbe*-UV-filters leads to more rigid formulations, with increased spreadability when compared to the control sample. As described in [41], above the yield stress value, the materials act as liquids, and thus, this value might be associated with the pourability of a formulation. As presented in Figure 6, BLN2-*Wbe*-UV-filers has the lowest yield stress value (~3 Pa), followed by BLN1-*Wbe*-UV-filers (~8 Pa), indicating that the addition of vegetable oil decreased the force needed to cause the formulation to flow [40].

Subsequently, frequency tests were performed within the previously established linear viscous region. The results indicated that all compositions exhibit a structured, solid-like behavior with a higher prominence of *G*′ over *G*″, nearly independent of frequency in the studied range (Figure 6a). Moreover, considering that no crossover of the G′ and G″ was registered, all compositions are considered to have a non-sticky nature. In view of the viscosity behavior, the complex viscosity (*η**) decreased with the increase in frequency (Figure 6b). Moreover, the rheological behavior was affected by the type of vegetable oil used to create the lipid matrix of BLN. As shown in Figure 6b, for the bioactive lipid nanocarriers prepared with *pomegranate oil* (BLN*2-Wbe*-UV-filters) has been determined the highest viscosity, while the lowest value of this parameter was recorded for the control sample.

For evaluation of the hydration effect, the two selected topical formulations based on BLN *willow extract* and UV-filters, were subjected to a rapid analysis of the moisture level of the skin.

The results obtained by applying the two hydrogels on the skin (Figure 7) showed the significant hydration properties manifested by the developed formulations. A markedly higher action was reported for the hydrogel containing BLN*2-Wbe*-UV-filters prepared with *pomegranate oil*. The latter shows a high degree of dehydration prevention.

## 4. Conclusions

The preparation of BLN-*Wbe/Ile*-UV-filters resulted in the development of efficient herbal nanostructured hydrogels with desirable rheological and controlled release features as well as photoprotective and hydration properties, which could significantly impact the pharmaco-cosmetic efficacy. BLN-based hydrogels exhibited impressive results in terms of photoprotective properties; for instance, the extremely reduced amount of UV filters incorporated into NLC-HG (e.g., 3.2% OCT and 2.26% BMDBM) have been able to combat about 90% of UV-B rays, simultaneously with an absorption of 98% of UVA rays. When OCT and BMDBM filters were incorporated together with *Wbe* into lipid nanocarriers, the *in vitro* skin permeability of UVA and UVB filters has been drastically reduced, which emphasizes a desired delay in skin permeability and a mainly UV-filters distribution on the skin surface. The major role of BLN-HG in improving the main features of active cosmetic ingredients leads to their outstanding application as efficient and healthful dermal carrier systems and may provide new alternative to existing skin-related products.

## Figures and Tables

**Figure 1 nanomaterials-12-02362-f001:**
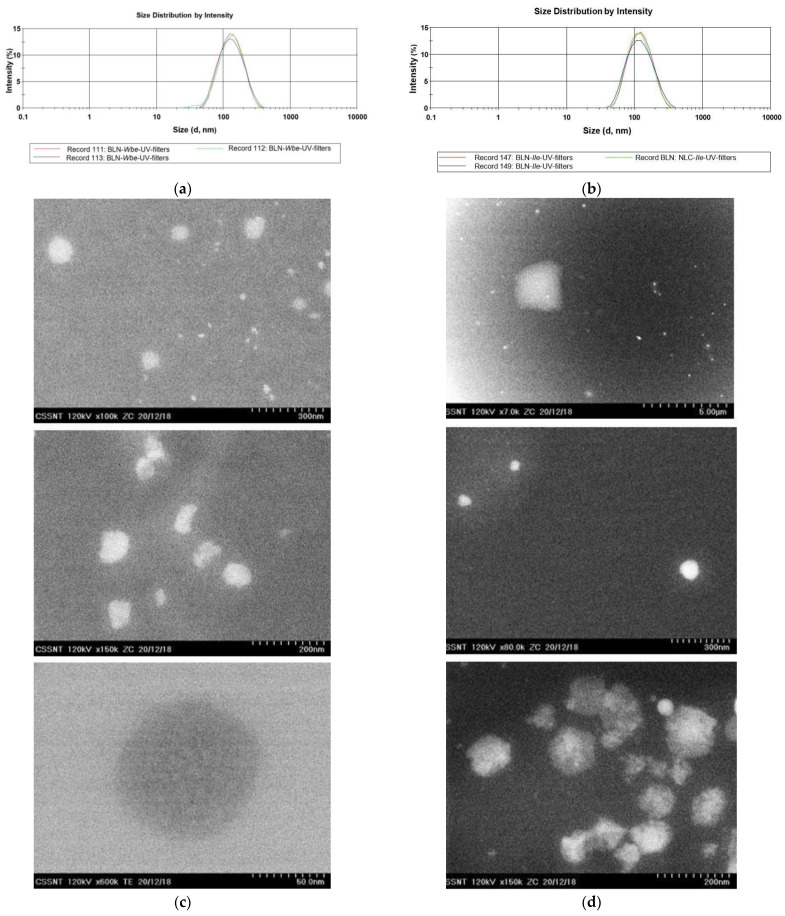
The particle size distribution (by DLS measurement) for BLN co-loaded with *Wbe* and UV-filters (**a**) and *Ile* and UV-filters (**b**). Morphology (by TEM) obtained for BLN co-loaded with UV-filters and herbal extract: *Wbe* (**c**) and *Ile* (**d**).

**Figure 2 nanomaterials-12-02362-f002:**
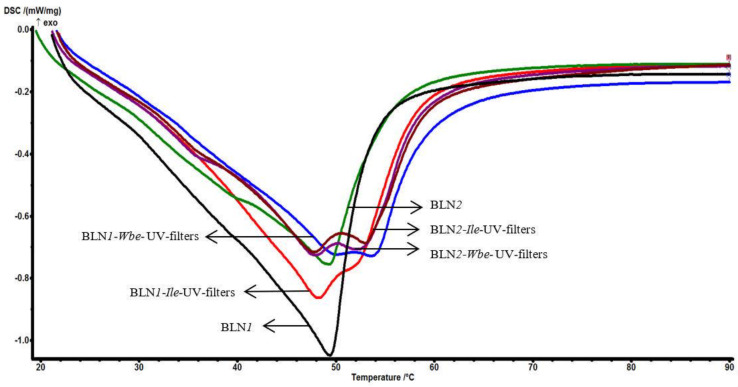
Differential scanning calorimetry for free-BLN and BLN co-loaded with herbal extract (*Ile*/*Wbe*) and UV-filters (OCT and BMDBM).

**Figure 3 nanomaterials-12-02362-f003:**
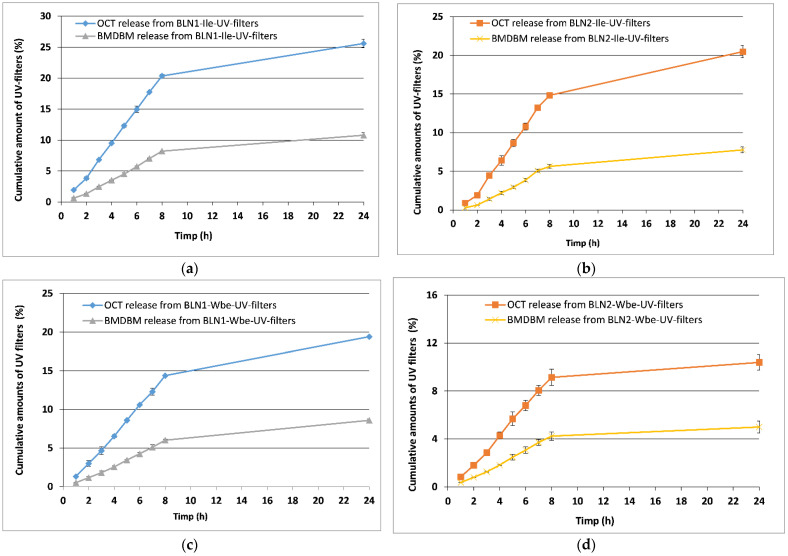
OCT and BMDBM release from BLN-*Ile*-UV-filters prepared with *carrot oil* (**a**) and *pomegranate oil* (**b**). OCT and BMDBM release from BLN-*Wbe*-UV-filters prepared with *carrot oil* (**c**) and *pomegranate oil* (**d**). Data are represented as mean ± SD, *n* = 3.

**Figure 4 nanomaterials-12-02362-f004:**
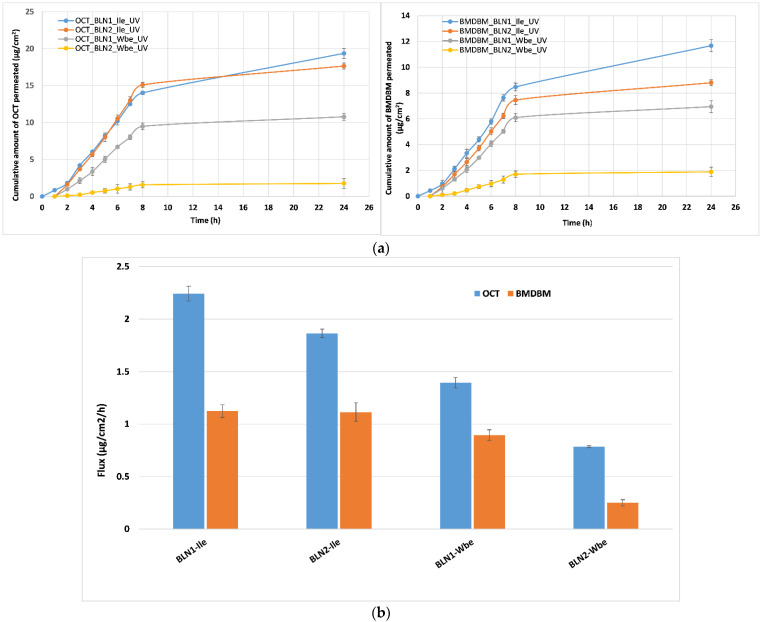
(**a**) *In vitro* permeation profiles of OCT and BMDBM from hydrogels with BLN-*Ile*/*Wbe*-UV-filters. (**b**) Fluxes at steady state of OCT and BMDBM filters from hydrogels with BLN- *Ile*/*Wbe*-UV-filters. Data are represented as mean ± SD, *n* = 3.

**Figure 5 nanomaterials-12-02362-f005:**
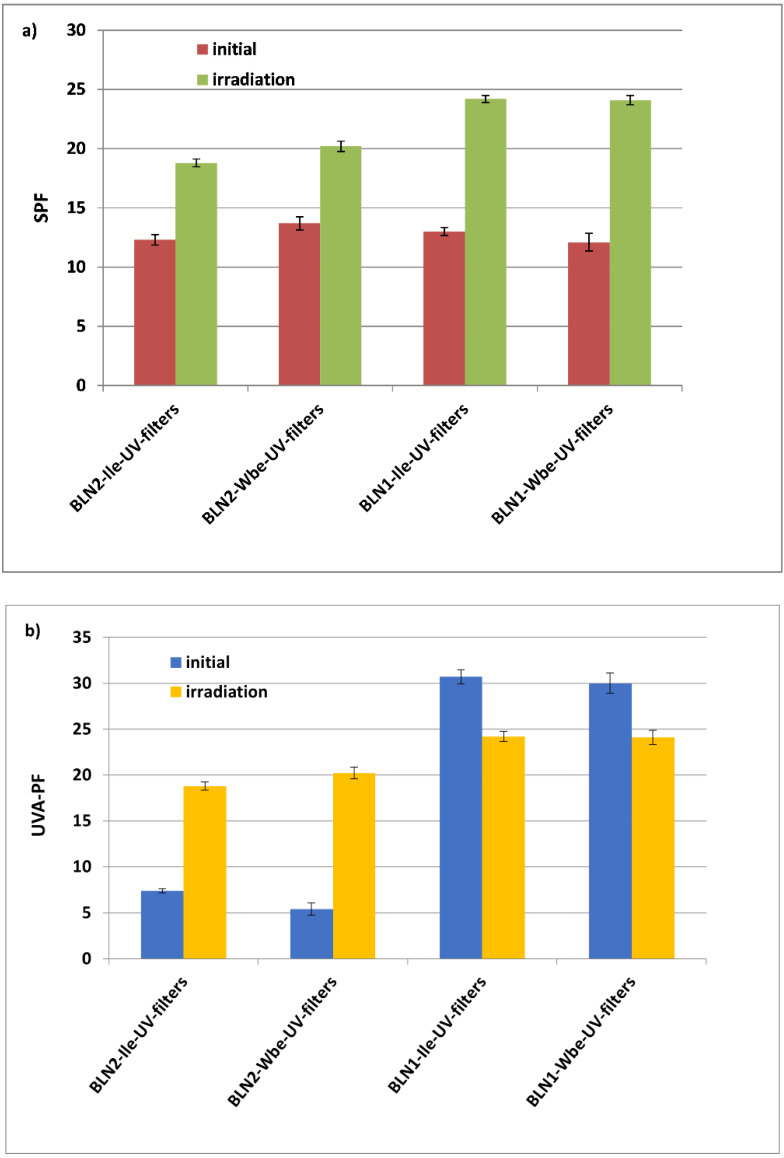
SPF and UVA-PF of hydrogels containing BLN-*Ile*/*Wbe*-UV-filters, SPF (**a**) and UVA-PF index (**b**). Data are represented as mean ± SD, *n* = 3.

**Figure 6 nanomaterials-12-02362-f006:**
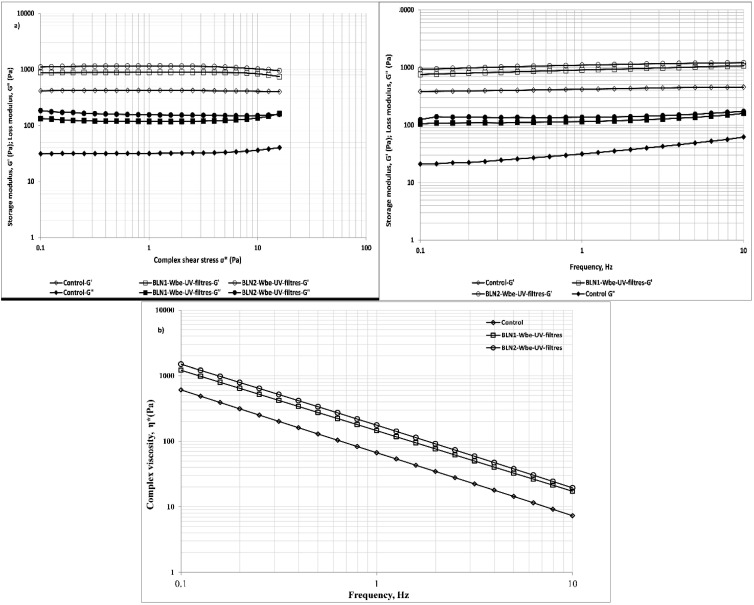
Rheological behavior of hydrogels containing BLN-*Wbe*-UV-filters: (**a**) The elastic modulus (*G*′, *Pa*) and the viscous (loss) modulus (*G*″, *Pa*) as a function of pressure and as a function of frequency; (**b**) The viscosity characteristics.

**Figure 7 nanomaterials-12-02362-f007:**
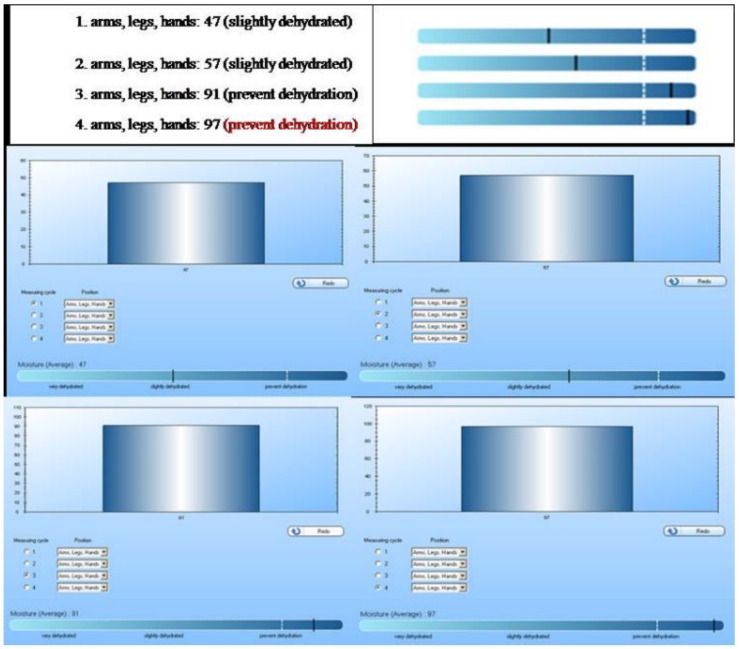
The hydration effect of BLN-*Wbe*-UV-filters: **1.** untreated skin (control); **2.** skin treated with free-HG; **3.** skin treated with HG based BLN*1*-*Wbe*-UV-filters; **4.** skin treated with HG based BLN*2*-*Wbe*-UV-filters.

**Table 1 nanomaterials-12-02362-t001:** Composition of BLN containing UV-filters and herbal extracts.

BLN Formulations	Surf. Mixture (g)	GM (g)	CP (g)	Vegetable Oil (g)	Herbal Extract (g)	UV-Filters * (g)
BLN*1*	2	3.5	3.5	3 *Co*	-	-
BLN*1*-*Ile*-UV-filters	2	3.5	3.5	3 *Co*	0.8	1.2
BLN*1*-*Wbe*-UV-filters	2	3.5	3.5	3 *Co*	0.8	1.2
BLN*2*	2	3.5	3.5	3 *Po*	-	-
BLN*2*-*Ile*-UV-filters	2	3.5	3.5	3 *Po*	0.8	1.2
BLN*2*-*Wbe*-UV-filters	2	3.5	3.5	3 *Po*	0.8	1.2

* The weight ratio between UVA (BMDBM) and UVB (OCT) filters was 1:1.4.

**Table 2 nanomaterials-12-02362-t002:** Physicochemical characteristics of bioactive lipid nanocarriers *.

Type of (BLN) **	Zave [nm]	PdI	*ξ* [mV]	EE%, *w*/*w*			DL%, *w*/*w*		
				*Ile*/*Wbe*	BMDBM	OCT	*Ile*/*Wbe*	BMDBM	OCT
BLN*1*	73.5 ± 0.72	0.138 ± 0.005	−32.9 ± 0.50	-	-	-	-	-	-
BLN*1*-*Ile*-UV-filters	120.2 ± 0.76	0.138 ± 0.010	−43.2 ± 2.17	80.1 ± 1.42	76.44 ± 0.53	57.83 ± 0.83	3.02 ± 0.08	3.70 ± 0.04	3.87 ± 0.05
BLN*1*-*Wbe*-UV-filters	110.8 ± 0.82	0.14 ± 0.005	−39.2 ± 0.61	82.3 ± 2.20	86.05 ± 1.76	74.79 ± 0.39	3.18 ± 0.12	4.18 ± 0.13	4.97 ± 0.09
BLN*2*	143.4 ± 1.82	0.232 ± 0.002	−34.8 ± 0.31	-	-	-	-	-	-
BLN*2*-*Ile*-UV-filters	171.3 ± 0.76	0.203 ± 0.008	−51.0 ± 0.82	89.2 ± 0.70	83.58 ± 0.79	71.80 ± 0.64	3.66 ± 0.03	4.06 ± 0.03	4.77 ± 0.06
BLN*2*-*Wbe*-UV-filters	155.6 ± 2.17	0.163 ± 0.004	−51.2 ± 1.56	90.8 ± 1.18	89.50 ± 0.45	84.38 ± 0.75	3.77 ± 0.08	4.34 ± 0.02	5.56 ± 0.07

* Data are represented as mean ± SD, *n* = 3. ** BLN*1* = bioactive lipid nanocarriers prepared with *Carrot oil (Co*); BLN*2* = bioactive lipid nanocarriers prepared with *Pomegranate oil (Po); Ile* = ivy leaves extract; *Wbe* = willow bark extract; UV-filters are butyl-methoxydibenzoylmethane (BMDBM) and octocrylene (OCT).

## Data Availability

Not applicable.

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
