# Peer review of "Salicin and Hederacoside C-Based Extracts and UV-Absorbers Co-Loaded into Bioactive Lipid Nanocarriers with Promoted Skin Antiaging and Hydrating Efficacy"

_nanomaterials, 2022, doi:10.3390/nano12142362_

Round 1
Reviewer 1 Report
You can find my comments in the text, as annotations. There is no need a major revision, just a minor one.
The novelty of the work should be highlighted in the manuscript.

Author Response
Dear Reviewers,
Thank you for the valuable comments and suggestions which help us in further improving of the manuscript “Herbal extract and UV-absorbers co-loaded into bioactive lipid nanocarriers with promoted skin antiaging and hydrating efficacy”. Based on the helpful Reviewer comments, we have carefully reviewed the manuscript.
All comments provided by the reviewer have been addressed and presented in the revised manuscript with a yellow background.
REVIEWER 1
- The novelty of the research should be highlited.
Author’s answer: the research novelty was introduced in the Abstract.
- Line 66-68 this part should be edited.
Author’s answer: on your recommendation, this phrase has been reissued.
- Please add some information in the text about main active ingredients of the extracts if it si possible.
Author’s answer: The two herbal extracts (supplied by S.C. Biopharm, Bucharest, Romania) are commercialized/marketed as standardized extracts in the main active ingredient: ivy leaves extract is standardized based on the content of Hederacoside C, e.g., 14.65% Hederacoside C; willow bark extract is standardised based on the content of Salicin, e.g., 10% Salicin.
- The methods used for statistical analysis should be mentioned in materials and method part
Author’s answer: In the materials and method section we added the statistical analysis which has been used.
- TEM for control nanocarriers (NLC-free);
Author’s answer: unfortunately, it was not possible (in time) to resynthesize and analyze TEM for free - bioactive nanocarriers
- Line 445 – maybe the use of glycerin instead of carrot or pomegranate oil helps to load more amounts of the extracts, as glycerin has more affinity to disolve water soluble compounds.
Author’s answer: Yes, you are right about improving the efficiency of Wbe/Ile encapsulation but replacing the two vegetable oils with glycerin will take away the benefits of carrot and pomegranate oils; on the other hand, the replacement of oils with glycerin could result in a decrease in the capture efficiency of the two UVA and UVB filters.
In summary, all the comments and suggestions provided by the Reviewers were taken into account and fully implemented. These are highlighted in the pdf document with yellow background, and we hope that they have made considerable improvements to the manuscript.
Best regards,
Prof. Nicoleta Badea

Reviewer 2 Report
Manuscript ID: jcs-1704912
Title: Herbal extract and UV-absorbers co-loaded into bioactive lipid nanocarriers with promoted skin antiaging and hydrating efficacy
The authors have prepared BLN-Wbe/Ile-UV-filters that resulted in herbal nanostructured hydrogels with desirable rheological and controlled release features for photoprotective and hydration properties, which may be useful for cosmetic applications.
Overall, the experiments were carefully conducted, and the manuscript was well organized. However, the manuscript should be revised according to the following comments:
Quoting a few examples below:
In the title: The tile should be modified since the terms of Herbal extract and UV-absorbers are too broad to cover this specific cosmetic DDS study.
In the Abstract:
The quotation mark should be placed properly.
The study will answer the question "how can willow bark extract (Wbe) or ivy leaf extract (Ile) influence the photoprotective, skin permeation and hydration properties of Bioactive Lipid Nanocarriers loaded with UV-filters and selected herbals.
In the main text:
Too many grammatical errors.
A space between a number and a unit of measurement (e.g. 75oC à 75 oC) should be given throughout the main text. Minor corrections are needed on putting commas in the sentences (i.e., A, B, and C), and formatting the references in MDPI format.
The quality of all the figures is very poor so hard to read. The excel format of all the plots should be modified with professional software (e.g. Prism). Please improve it thoroughly.
Although the authors demonstrated the physicochemical activity and rheological properties of the nanocarrier products, there is no benchmarkable or comparable information with previous literatures, which can make the findings reasonable and publishable on Nanomaterials.
How about quantitative DDS performance such as loading and release comparison of this system with other references elsewhere? A quantitative analysis with a reference would be highly recommended to increase the quality of the paper.
Author Response
Dear Reviewers,
Thank you for the valuable comments and suggestions which help us in further improving of the manuscript “Herbal extract and UV-absorbers co-loaded into bioactive lipid nanocarriers with promoted skin antiaging and hydrating efficacy”. Based on the helpful Reviewer comments, we have carefully reviewed the manuscript.
All comments provided by the reviewer have been addressed and presented in the revised manuscript with a yellow background.
Title: Herbal extract and UV-absorbers co-loaded into bioactive lipid nanocarriers with promoted skin antiaging and hydrating efficacy
The authors have prepared BLN-Wbe/Ile-UV-filters that resulted in herbal nanostructured hydrogels with desirable rheological and controlled release features for photoprotective and hydration properties, which may be useful for cosmetic applications.
Overall, the experiments were carefully conducted, and the manuscript was well organized. However, the manuscript should be revised according to the following comments:
Quoting a few examples below:
In the title: The tile should be modified since the terms of Herbal extract and UV-absorbers are too broad to cover this specific cosmetic DDS study.
Author’s answer: the title has been changed in “Salicin and Hederacoside C-based extracts and UV-absorbers co-loaded into bioactive lipid nanocarriers with promoted skin antiaging and hydrating efficacy”
In the Abstract:
The quotation mark should be placed properly.
The study will answer the question "how can willow bark extract (Wbe) or ivy leaf extract (Ile) influence the photoprotective, skin permeation and hydration properties of Bioactive Lipid Nanocarriers loaded with UV-filters and selected herbals.
Author’s answer: It's done
In the main text: Too many grammatical errors.
Minor corrections are needed on putting commas in the sentences (i.e., A, B, and C), and formatting the references in MDPI format.
Author’s answer: These corrections have been made in the entire manuscript.
The quality of all the figures is very poor so hard to read. The excel format of all the plots should be modified with professional software (e.g. Prism). Please improve it thoroughly.
Author’s answer: We have improved the quality of the figures. The zip archive on the platform contains the figures at a suitable resolution.
Although the authors demonstrated the physicochemical activity and rheological properties of the nanocarrier products, there is no benchmarkable or comparable information with previous literatures, which can make the findings reasonable and publishable on Nanomaterials. How about quantitative DDS performance such as loading and release comparison of this system with other references elsewhere? A quantitative analysis with a reference would be highly recommended to increase the quality of the paper.
Author’s answer: Indeed, the reviewer is right. For improving the manuscript information, we have compared previous literature with our result. These comparisons mainly focused on the behaviour of the two sunscreens in other related nanostructured lipid carriers (e.g., physico-chemical features, photoprotection, photostability and rheological behaviour). A deeply references comparison of both active ingredients – UV-absorbers and plant extracts, willow bark extract (Wbe) or ivy leaf extract (Ile) – was not possible because no studies on the co-optation of plant extracts (Ile and Wbe) with OCT and BMDBM filters in the same nanostructured distribution system were reported.
Best regards,
Prof. Nicoleta Badea

Reviewer 3 Report
In the paper ‘ Herbal extract and UV-absorbers co-loaded into bioactive lipid nanocarriers with promoted skin antiaging and hydrating efficacy‘ authors investigated the effect of willow bark extract and ivy leaf extract on the photoprotective, skin-permeable and moisturizing properties of Bioactive Lipid Nanocarriers filled with UV filters and selected herbs. The obtained products were characterized for particle size, zeta potential, thermal behavior, entrapment efficiency and drug loading. Authors found out that the obtained creams / products showed the desired rheological properties along with the controlled release as well as photo-protective and hydration properties, which could significantly affect the pharmaco-cosmetic effectiveness. However, in my opinion, the paper requires significant changes. Below are my comments:
· Introduction - too long, shorten it and focus on the most important issues.
· Methods: Written rather chaotically, the authors refer to their previous work, but in my opinion the general procedure is meant to be understandable to the reader.
· In the text the authors use the abbreviations UVA, AVB and UV-A, UV-B in many places. You have to be consistent and choose one correct version.
· The literature review contains 38 references, of which 33 are in the introduction. It can be said that there are no references in the part of discussing the results. In my opinion, the obtained results should be compared with the literature data, unless it is a novelty on a global scale and no one has done anything similar so far.
· Line 364: ‘The size and morphology characterization of the lipid bio-nanocarriers revealed the obtaining of delivery systems with average diameters between 73 and 171 nm’. I do not know where the authors took the size of 73 nm from, it cannot be seen in DLS or in TEM micrographs. The microscopic photos are quite vague and it is rather difficult to notice any capsules that the authors write about all the time.
To sum up, I believe that the work in this form is not suitable for publication.

Author Response
Dear Reviewers,
Thank you for the valuable comments and suggestions which help us in further improving of the manuscript “Herbal extract and UV-absorbers co-loaded into bioactive lipid nanocarriers with promoted skin antiaging and hydrating efficacy”. Based on the helpful Reviewer comments, we have carefully reviewed the manuscript.
All comments provided by the reviewer have been addressed and presented in the revised manuscript with a yellow background.
In the paper ‘Herbal extract and UV-absorbers co-loaded into bioactive lipid nanocarriers with promoted skin antiaging and hydrating efficacy‘ authors investigated the effect of willow bark extract and ivy leaf extract on the photoprotective, skin-permeable and moisturizing properties of Bioactive Lipid Nanocarriers filled with UV filters and selected herbs. The obtained products were characterized for particle size, zeta potential, thermal behavior, entrapment efficiency and drug loading. Authors found out that the obtained creams/products showed the desired rheological properties along with the controlled release as well as photo-protective and hydration properties, which could significantly affect the pharmaco-cosmetic effectiveness. However, in my opinion, the paper requires significant changes. Below are my comments:
- Introduction - too long, shorten it and focus on the most important issues.
Author’s answer: The information in the Introduction section have been reorganized and the references were reduced.
- Methods: written rather chaotically, the authors refer to their previous work, but in my opinion the general procedure is meant to be understandable to the reader.
Author’s answer: According to the recommendation, the experimental methodology has been reorganized in some places to make it easier to understand.
- In the text the authors use the abbreviations UVA, AVB and UV-A, UV-B in many places. You have to be consistent and choose one correct version.
Author’s answer: we corrected in entire paper: UVA and UVB
- The literature review contains 38 references, of which 33 are in the introduction. It can be said that there are no references in the part of discussing the results. In my opinion, the obtained results should be compared with the literature data, unless it is a novelty on a global scale and no one has done anything similar so far.
Author’s answer: As recommended by the reviewer, we have included some comparative results from related literature studies. Following the reorganization of the Introduction section (which now covers 28 references), 10 new references have been included in the "Results and Discussions" section, in order to compare our results with the literature data.
Line 364: ‘The size and morphology characterization of the lipid bio-nanocarriers revealed the obtaining of delivery systems with average diameters between 73 and 171 nm’. I do not know where the authors took the size of 73 nm from, it cannot be seen in DLS or in TEM micrographs. The microscopic photos are quite vague and it is rather difficult to notice any capsules that the authors write about all the time.
Author’s answer: The specified size presented in Table 2 (e.g., ranged between 73 and 171 nm) represents the mean average diameters/Zave determined statistically by DLS analysis (Fig. 1 a, b). This majority lipid population smaller than 200 nm can be also seen by viewing TEM images – Fig. 1 c, d (200 or 300 nm scale).
Best regards,
Prof. Nicoleta Badea

Round 2
Reviewer 3 Report
After the changes made I believe that the work in this form is suitable for publication
Author Response
Dear Reviewer,
Thank you for your appreciation.
Yours sincerely,
Prof. dr. Nicoleta Badea